# Wearable Technologies for Electrodermal and Cardiac Activity Measurements: A Comparison between Fitbit Sense, Empatica E4 and Shimmer GSR3+

**DOI:** 10.3390/s23135847

**Published:** 2023-06-23

**Authors:** Vincenzo Ronca, Ana C. Martinez-Levy, Alessia Vozzi, Andrea Giorgi, Pietro Aricò, Rossella Capotorto, Gianluca Borghini, Fabio Babiloni, Gianluca Di Flumeri

**Affiliations:** 1Department of Computer, Control, and Management Engineering, Sapienza University of Rome, 00185 Rome, Italy; pietro.arico@uniroma1.it (P.A.); capotorto.1843967@studenti.uniroma1.it (R.C.); 2BrainSigns Srl, 00198 Rome, Italy; ana.martinez@uniroma1.it (A.C.M.-L.); alessia.vozzi@uniroma1.it (A.V.); andrea.giorgi@uniroma1.it (A.G.); gianluca.borghini@uniroma1.it (G.B.); fabio.babiloni@uniroma1.it (F.B.); gianluca.diflumeri@uniroma1.it (G.D.F.); 3Department of Molecular Medicine, Sapienza University of Rome, 00185 Rome, Italy; 4Department of Anatomical, Histological, Forensic and Orthopaedic Sciences, Sapienza University of Rome, 00185 Rome, Italy; 5College of Computer Science and Technology, Hangzhou Dianzi University, Hangzhou 310005, China

**Keywords:** electrodermal activity, photoplethysmography, consumer wearables

## Abstract

The capability of measuring specific neurophysiological and autonomic parameters plays a crucial role in the objective evaluation of a human’s mental and emotional states. These human aspects are commonly known in the scientific literature to be involved in a wide range of processes, such as stress and arousal. These aspects represent a relevant factor especially in real and operational environments. Neurophysiological autonomic parameters, such as Electrodermal Activity (EDA) and Photoplethysmographic data (PPG), have been usually investigated through research-graded devices, therefore resulting in a high degree of invasiveness, which could negatively interfere with the monitored user’s activity. For such a reason, in the last decade, recent consumer-grade wearable devices, usually designed for fitness-tracking purposes, are receiving increasing attention from the scientific community, and are characterized by a higher comfort, ease of use and, therefore, by a higher compatibility with daily-life environments. The present preliminary study was aimed at assessing the reliability of a consumer wearable device, i.e., the Fitbit Sense, with respect to a research-graded wearable, i.e., the Empatica E4 wristband, and a laboratory device, i.e., the Shimmer GSR3+. EDA and PPG data were collected among 12 participants while they performed multiple resting conditions. The results demonstrated that the EDA- and PPG-derived features computed through the wearable and research devices were positively and significantly correlated, while the reliability of the consumer device was significantly lower.

## 1. Introduction

Nowadays, wearable technologies are constantly growing in terms of popularity, and wearable devices are increasingly employed in monitoring fitness and health-related parameters [1,2]. Wearable technology relies on tiny, accurate, and cost-efficient sensors, and it can be used to track people’s movement, collect biometric signals, and analyse daily activities [2,3,4,5]. One of the most popular products in terms of consumer-grade wearables is the wristband. Such a technology is fully compatible with real-world applications, and supported by a very low grade of invasiveness, which does not negatively interfere with the users’ activities, by a consistent battery life, and wireless connectivity to fetch and stream data [6,7,8].

Therefore, the scientific community has also largely started to investigate the reliability of such wearable technologies in evaluating neurophysiological and autonomic parameters. In this context, several scientific works demonstrated the reliability of wearable devices, such as the Empatica E4, in collecting Electrodermal Activity (EDA) and Photoplethysmographic (PPG) signals [9,10,11,12]. As demonstrated by different previous works [4,10,13,14,15], the EDA corresponds to a consistent biomarker of the stress level, while the PPG-derived parameters, such as the Heart Rate (HR) and the Heart Rate Variability (HRV), were significantly correlated to the mental workload and emotional state variations [16,17,18,19,20,21]. In particular, Ragot et al. [19] successfully adopted the Empatica E4 wristband to measure physiological responses in an emotion recognition task. Based on this evidence, wearable devices were also used to assess different mental states, such as in the study conducted by Setz et al. [14], in which mental drowsiness was detected during a driving simulation task through HRV estimation. In this context, other studies have demonstrated the capability of wearable devices in detecting crucial mental states in operational environments, such as the prediction of the stress and the mental workload in police academy training [22] and during professional physical activity [23] through the PPG, respiration, and motion-activity data analyses. More recently, Giorgi et al. [9] positively assessed the reliability of the Empatica E4 in detecting EDA and cardiac activity with respect to regular laboratory equipment, considered the gold standard in terms of accuracy.

To summarize, wearable technologies for research, e.g., the Empatica E4, have been shown to be reliable in evaluating neurophysiological and autonomic parameters. However, they are still characterized by a relatively non-negligible economic cost (i.e., between 1 and 2 thousand EUR/USD), that make them less compatible with massive use, such as large-scale studies involving multiple participants at the same time. Therefore, further developments could be made using consumer wearable technologies. In this regard, the latest technological developments allowed on the release of devices in consumer markets able to capture autonomic parameters, such as PPG and EDA data. The Fitbit Sense was one of the first consumer wearables capable of detecting such signals.

The reliability of such a device in detecting autonomic and health-related parameters compared to research wearable and laboratory devices may play a crucial role, due to its relevantly high economic accessibility. In this regard, it must be considered that a more accessible device could be more widely employed in scientific research for applying methodologies relying on neurophysiological and autonomic signal analysis. Nevertheless, there is still a consistent lack of scientific works to investigate the reliability of a consumer wearable compared to the accuracy of research wearables and laboratory equipment. In particular, to the best of our knowledge, there is not any study so far aiming at investigating the reliability of a consumer-grade wearable smartband, i.e., the Fitbit, in measuring neurophysiological autonomic data, namely PPG and EDA, with respect to research-grade devices. In this regard, there have been several notable works related to the employment of the most recent Apple Watch generations (Apple Inc., Cupertino, CA, USA) in medical-graded experimental protocols, with promising results [24,25,26]. In particular, the consumer-grade wearable was employed for investigating its capabilities in detecting cardiac-related pathologies, such as atrial fibrillation ([27]), and its capability in assessing noise levels within an intensive care unit, as proposed by Scquizzato and colleagues ([28]). In this context, it has to be noted that the reliability of the above-mentioned consumer-grade wearable, i.e., the Apple Watch, was not investigated in terms of collecting neurophysiological and autonomic data, such as PPG and EDA.

Therefore, the aim of the present study consisted of assessing the reliability of the Fitbit Sense, through the estimation of EDA- and PPG-derived parameters compared to a research wearable and a laboratory device, i.e., the Empatica E4 and the Shimmer GSR3+, respectively. Because of the difficulty in recruiting participants, given the fact that the complete experimental protocol was designed and conducted within the COVID-19 pandemic restrictions, the sample size included in the experimental protocol was low (12 participants). Therefore, the presented work is intended as a preliminary study; however, it has to be considered relevant in any case because of the novelty of the topic, it being the first study investigating a consumer-grade device’s abilities to collect reliable data for scientific purposes.

## 2. Materials and Methods

Twelve (12) participants, young students (18–26 years old, 6 males and 6 females) from the Sapienza University of Rome, with normal or corrected-to-normal vision, were recruited on a voluntary basis. Informed consent was obtained from each participant after explanation of the study. The experiment was conducted following the principles outlined in the Declaration of Helsinki of 1975, as revised in 2000 and was approved by the Sapienza University of Rome Ethical Committee in Charge for the Department of Molecular Medicine (protocol number: 2104/2021, approved on 5 April 2021). To respect the privacy of participants, only aggregate results were reported, and any results based on single identity analysis was presented.

The experimental procedure was designed as simple as possible, in order to allow a consistent reliability assessment of the three kinds of devices involved, i.e., the consumer wearable Fitbit Sense (Google Inc., Mountain View, CA, USA), the research wearable Empatica E4 (Empatica, Milan, Italy), and the laboratory equipment Shimmer GSR3+ (Shimmer Sensing, Dublin, Ireland) (Figure 1). The experimental tasks corresponded to three consecutive 90 s long resting conditions, a minimum requirement for collecting valid data from the Fitbit Sense. On a 90 s long recording session, the Fitbit provides its estimation of EDA responses and HR every 30 s. In this way, for each participant, 9 data points were available. All the participants involved in the experimental protocol were instructed to not perform any activities during the neurophysiological and autonomic data collection along the entire experiment.

### 2.1. PPG Signal Recording and Analysis

The PPG signal was collected simultaneously through the Empatica E4 and the Shimmer GSR3+. The first one was placed on the participants’ right wrist, the Empatica E4 on the participants’ left wrist, while the Shimmer GSR3+ PPG sensor was placed on the participants’ thumb (Figure 2). Firstly, the PPG signals were filtered using a 5th Butterworth band-pass filter (1–15 Hz) in order to reject the continuous component and the high-frequency interferences, such as that related to the main power source. Subsequently, a signal preprocessing chain was applied to identify and remove movement-based artifacts. In particular:Signal filtering between 1 and 15 Hz.All the negative signal segments were made positive by checking their skewness.Signal windowing according to a 2 s length. This was based on the hypothesis that, along 2 s, at least one physiological R-peak must occur.The local maxima of each 2 s long window were considered.The identification of artefactual portions through a threshold method. This step was performed through specific sub-steps:
○Each 2 s long window was labelled as an artefact if the corresponding local maximum exceeded the threshold of 2 * median amplitude among all the local maxima. This corresponded to an empiric and reasonable assumption for which if a signal peak exceeds twice the median among the other local maxima it corresponds to a non-physiological peak.○Each 2 s long window was labelled as an artefact if the corresponding local maximum did not exceed a minimum threshold set around 0. This corresponded to an empirical and reasonable assumption for which if a signal peak does not exceed a minimum threshold around 0 it corresponds to a non-physiological peak or, more likely, it corresponds to a 2 s long window in which no signal was recorded.
The identification of artefactual portions through a threshold method and the first signal derivative. This step was based on the assumption for which specific signal artefacts, e.g., the signal discontinuities, are characterized by an amplitude within a physiological range and, therefore, they are not identifiable via the previous threshold method. In this regard, the signal first derivative was considered to identify non-physiological discontinuities. This step was performed through specific sub-steps:
○The signal first derivative was standardized and squared.○The convolution between the signal first derivative standardized and squared and a 0.15 s long time window.○Each 2 s long window was labelled as an artefact if the first signal derivative standardized and squared exceeded the threshold of 9. This value ensured that the labelled signal portion corresponded to a physiological outlier for the 99.7%, corresponding to 3 * σ according to the cumulative distribution. This corresponded to an empirical and reasonable assumption for which if the signal first derivative exceeds such a threshold it corresponds to a non-physiological signal discontinuity.


Subsequently, the Pan–Tompkins [9,29,30] algorithm was applied exclusively to the non-artefactual PPG signal portions for the Inter-Beat Interval (IBI) estimation. Finally, the HR parameter was estimated from the IBI during the task length for each participant, and averaged on 30 s long windows according to the measures provided by the Fitbit device.

Alternatively, since it is not possible to access the Fitbit raw data, the HR estimation computed by the Fitbit software (and related application) was considered during the same task.

### 2.2. EDA Recording and Analysis

Similarly to the PPG signal, the EDA was collected through all three devices. Actually, also in this case, the Fitbit device does not provide access to the raw data, so the Shimmer and Empatica data were recorded and processed as follows, while the Fitbit output was considered for the analysis as described in the next sub-paragraph. The sampling frequency of the Shimmer3 GSR+ unit laboratory device corresponded to 64 Hz while the sampling frequency of the Empatica E4 was 4 Hz. Shimmer sensors were placed on the participant’s non-dominant hand on the second and third fingers. Concerning the Empatica E4, the two electrodes were placed on the bottom part of the wrist, while the Fitbit Sense required the physical contact between the participants’ hand palm and the upper part of the device in order to collect EDA. Regarding the signal processing related to the EDA collected through the Shimmer GSR3+ and the Empatica E4, the signal was firstly low-pass filtered with a cut-off frequency of 1 Hz and then processed using the Ledalab suite [31], a specific open-source toolbox implemented within the MATLAB (MathWorks, Natik, Massachussets) environment for EDA processing. The continuous decomposition analysis [32] was applied in order to estimate the tonic (SCL) and the phasic (SCR) EDA components [33]. The SCL corresponds to the slow-changing component of the EDA signal, consistently related to arousal and stress levels. On the contrary, the SCR is the fast-changing component of the EDA signal, usually related to single stimuli reactions. Finally, only the SCL was analysed accordingly, with previous scientific evidence suggesting this parameter is strictly correlated with specific and relevant mental states, such as stress [34,35,36]. The SCL estimated through the research wearable and laboratory equipment was computed during the task length and averaged in 30 s long windows according to the measures provided by the Fitbit device.

The Fitbit Sense does not allow continuous EDA raw data collection and access. The consumer wearable exclusively allowed us to collect the EDA responses, which consists of a parameter related to the Skin Conductance Level (SCL) peaks overtime. Therefore, the “SCL responses” measure provided each 30 s was considered.

For the final purpose of the presented study, any data normalization was required since the statistical analysis was based on the comparison along each data subject distribution. Therefore, any intra-subject differences were required to be removed.

### 2.3. Statistical Analysis

Given the main objective of the presented study, different types of correlation analyses were performed in order to compare the reliability of the consumer wearable to the research wearable and laboratory equipment. In particular, Pearson’s correlation analysis was performed to assess the similarity between the measurements collected through the consumer wearables, research wearables, and the laboratory equipment. Subsequently, the repeated measures correlation analysis [37] was additionally performed to estimate the reliability of the parameters estimated by the consumer and research wearable devices with respect to the laboratory one, both at the single-participant level and in the entire group, i.e., by taking into account intra- and inter-individual variability at the same time.

## 3. Results

Regarding the functional comparison between the consumer and research wearable devices and the laboratory equipment, the Pearson correlation analysis revealed a consistent difference between the measurements collected through the consumer wearable and the research wearable, e.g., the Fitbit Sense and the Empatica E4, respectively, compared to the laboratory equipment, e.g., the Shimmer GSR3+, considered the gold standard. In fact, Pearson’s correlation analysis performed on the PPG-derived parameter evaluations revealed a positive and significant correlation between the HR computed by the signal collected through the Empatica E4 and the Shimmer GSR3+ for a large majority of the participants (9/12) (Figure 3). No significant correlation for any participants was observed between the HR estimated through the Fitbit Sense and the Shimmer GSR3+.

Similarly, Figure 4 shows how the SCL parameters derived by the EDA collected through the Empatica E4 and the Shimmer GSR3+ were positively (all R > 0.4) and significantly (all *p* < 0.05) correlated for nearly all the participants (10/12), while the SCL parameters derived by the EDA gathered through the Fitbit Sense and the Shimmer GSR3+ revealed a positive (all R > 0.53) and significant (all *p* < 0.05) correlation only for 2 participants among 12.

Regarding the device reliability assessment at the single-participant level between the Empatica E4 and the Shimmer GSR3+, the repeated measure correlation analysis revealed a positive and significant correlation for both the SCL and HR parameters (SCL: R = 0.681, *p* < 10^−34^; HR: R = 0.596, *p* < 10^−33^) (Figure 5 and Figure 6).

However, the comparison between the Fitbit Sense and the Shimmer GSR3+ revealed a positive and significant correlation only for the SCL estimations (R = 0.412, *p* < 10^−25^) (Figure 7). In fact, positive and significant correlations were observed between the HR estimations provided by the Fitbit Sense, and one evaluated by the Shimmer GSR3+ (R = −0.121; *p* = 0.1) (Figure 8).

## 4. Discussion

The objective of the presented study was the preliminary assessment of the consumer and research wearables’ reliability compared to the laboratory equipment, considered the gold standard, in collecting PPG signal and EDA in order to further and reliably estimate autonomic parameters, such as the HR and SCL. This aspect is of relevance, since autonomic parameters can be associated with the objective evaluation of specific mental states [10,38,39,40]. Being able to estimate these parameters under real conditions would open up the possibility of reliably assessing mental states in operational environments where particularly invasive instrumentation, such as laboratory equipment, cannot be used [41].

The presented results have consistently demonstrated, accordingly to previous scientific evidences [2,8,9,14,42], the reliability of the research wearable, i.e., the Empatica E4, with respect to the Shimmer GSR3+. In fact, both Pearson’s correlation (Figure 3 and Figure 4) and the repeated measure correlation analyses (Figure 5 and Figure 6) revealed that for the great majority of participants, at least for 9 participants among 12, the HR and SCL parameters estimated through the signals collected by the Empatica E4 were highly (all R > 0.58) and significantly (all *p* < 0.05) correlated with the ones estimated from the signals collected through the Shimmer GSR3+. On the contrary, the consumer wearable, i.e., the Fitbit Sense, did not show the same grade of reliability. In particular, the correlation analyses showed positive and significant correlations for a very small participant number. The HR estimated through the Fitbit Sense did not significantly correlate with the one estimated through the Shimmer GSR3+. A different discussion needs to be had with regard to the SCL estimation through the Fitbit Sense. Even if Pearson’s correlation analysis highlighted a positive (all R > 0.48) and significant (all *p* < 0.05) correlation only for 2 participants among 12 (Figure 3), it has to be noted that the repeated measures correlation among all the participants was positive and significant (Figure 7). This result reflects a promising starting point to further investigate the consumer wearables’ reliability in the scientific research field.

From a mere methodological and technical point of view, it is important to note that, actually, the low reliability of Fitbit measures can be caused by the sensors or by the integrated software used for metric computation. In fact, the Fitbit does not provide access to the raw signal recorded, that is the EDA and the PPG, but only to the parameters related to these two neurophysiological activities computed by their proprietary software, respectively, the EDA responses and Heart Rate. This means that despite these results, it cannot said that by improving data processing and computation this consumer-grade device would increase its potential for this kind of application. At any rate, from a practical perspective the restricted access to raw data with third-party applications is a great limitation for scientific applications, where there is usually the need for data synchronization with other additional data, and for raw data access in order to investigate different data features and patterns as well as new procedures and algorithms for data processing. This precludes the possibility of applying techniques for identifying and correcting movement-based signal artifacts, which are very common in real-world applications and which might significantly and negatively interfere with the correct estimation of the autonomic parameters of interest. In the scientific literature, a plethora of works were already presented for validating data preprocessing techniques specifically developed for the PPG signal and EDA analyses in real-world applications [30,43,44].

As introduced, the present study is limited by the low sample size and probably by the novelty of the investigated consumer-grade device (the Fitbit Sense was just released on the market); therefore, new studies on this topic are strongly encouraged in order to verify, with larger sample sizes, whether technical improvements would have increased the reliability of these devices.

## 5. Conclusions

To conclude, the presented preliminary study demonstrated the consistent reliability of the usage of research wearables, i.e., the Empatica E4, in scientific research for real-world applications. Regarding consumer wearables, it must be observed that, at least for the EDA-derived parameter, the measurements were positively correlated at the single-participant level with the ones provided by the gold-standard equipment. This aspect leads to the conclusion that the reliability of consumer wearables must be further investigated, especially by combining raw data collection with specific preprocessing techniques. As highlighted in the Results section, the non-positive correlation observed between HR evaluated through the consumer-grade wearable device, i.e., the Fitbit Sense, and the ones estimated through the device considered the gold standard, i.e., the Shimmer GSR3+, could, relevantly, be related to the lack of application of ad hoc preprocessing techniques for the PPG raw signal collected through the Fitbit Sense. In this regard, it must be observed that consumer wearables represent the perfect trade-off between usability, since they can be used in real-world applications without negatively interfering with the user’s activities, and cost, given their very low economic cost compared to research wearables and laboratory equipment. Filling this gap in knowledge highlighted by the present work will significantly enlarge the possibility of employing consumer-grade wearable devices, such as the Fitbit Sense, in scientific research for longitudinally assessing relevant human mental states.

## Figures and Tables

**Figure 1 sensors-23-05847-f001:**
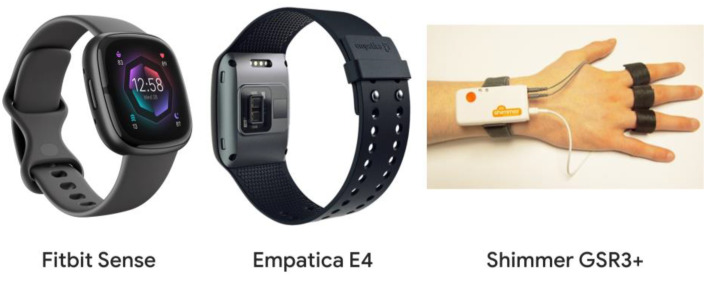
Overview of the three different devices type investigated in the present work.

**Figure 2 sensors-23-05847-f002:**
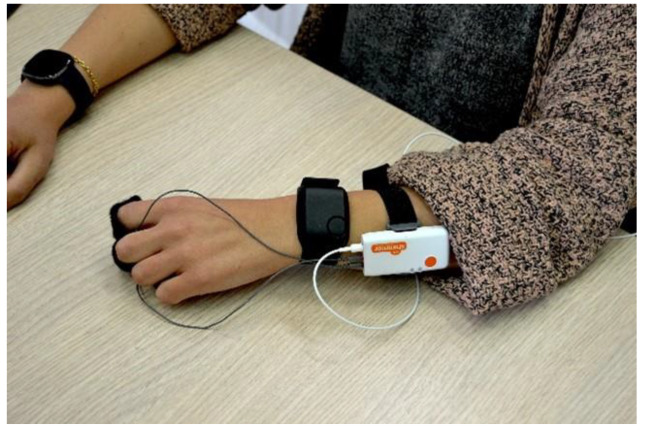
Overview of the experimental settings.

**Figure 3 sensors-23-05847-f003:**
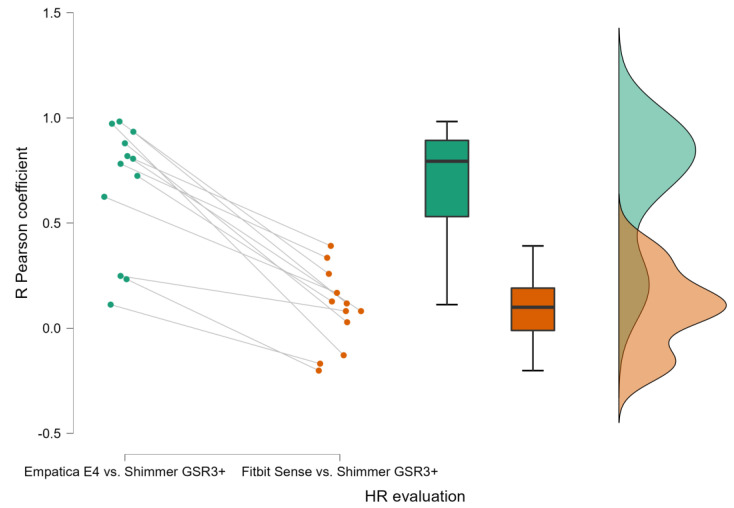
Person’s correlations between the Empatica E4 and Shimmer (**left bar**) and the Fitbit Sense and the Shimmer (**right bar**) in terms of Heart Rate estimation.

**Figure 4 sensors-23-05847-f004:**
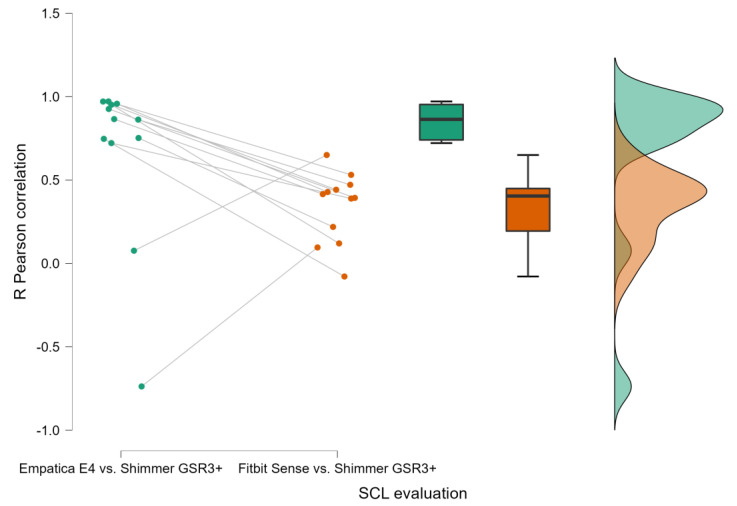
Person’s correlations between the Empatica E4 and Shimmer GSR3+ (**left bar**) and the Fitbit Sense and the Shimmer GSR3+ (**right bar**) in terms of Skin Conductance Level estimation.

**Figure 5 sensors-23-05847-f005:**
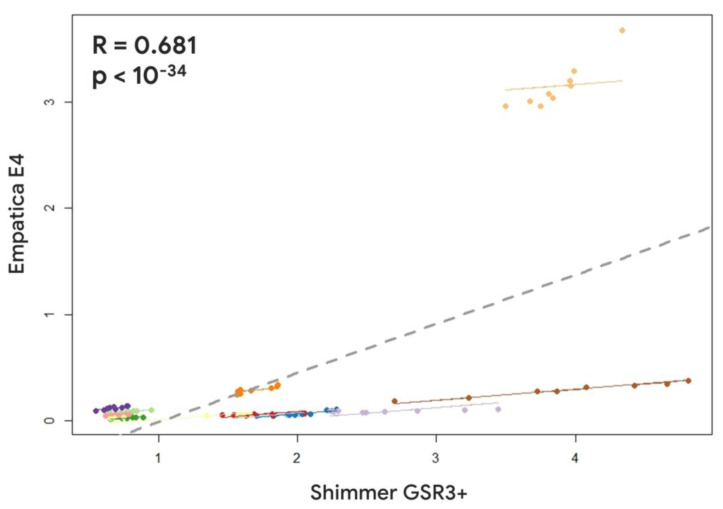
Repeated measures correlation between the Skin Conductance Level estimated through the Empatica E4 and the Shimmer GSR3+. Lines and dots characterized by the same colour represent the data distribution per each participant.

**Figure 6 sensors-23-05847-f006:**
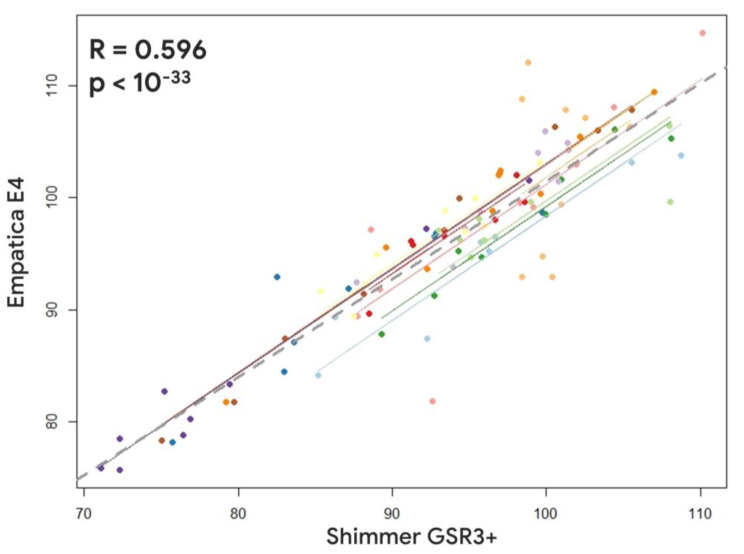
Repeated measures correlation between the Heart Rate estimated through the Empatica E4 and the Shimmer GSR3+. Lines and dots characterized by the same colour represent the data distribution per each participant.

**Figure 7 sensors-23-05847-f007:**
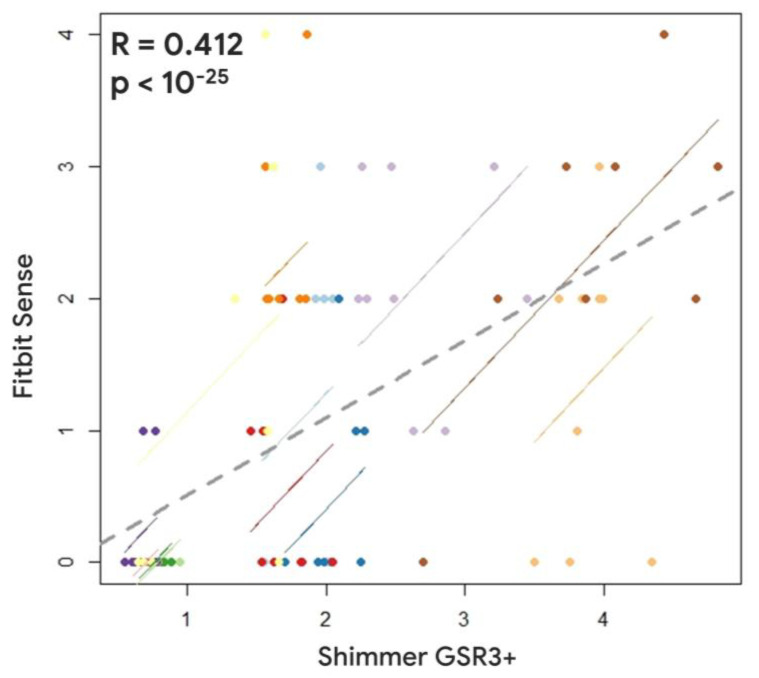
Repeated measures correlation between the Skin Conductance Level estimated through the Fitbit Sense and the Shimmer GSR3+. Lines and dots characterized by the same colour represent the data distribution per each participant.

**Figure 8 sensors-23-05847-f008:**
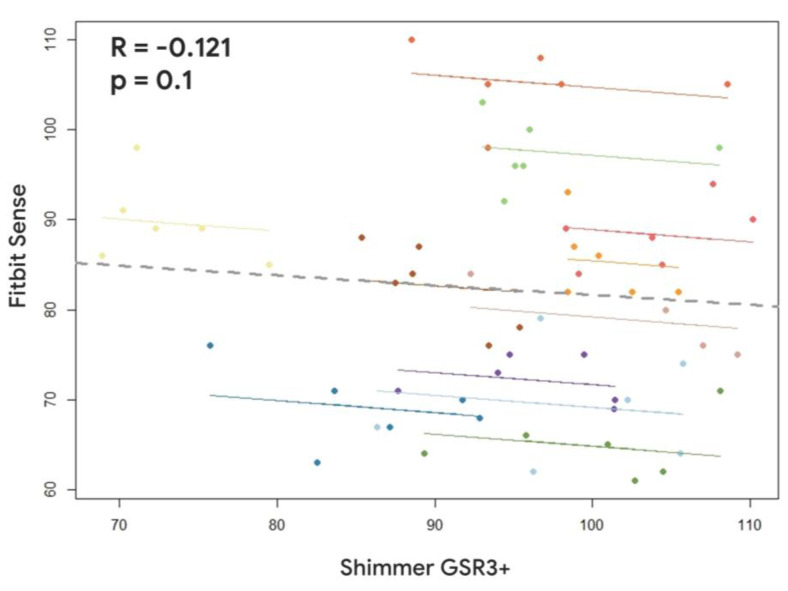
Repeated measures correlation between the Heart Rate estimated through the Fitbit Sense and the Shimmer GSR3+. Lines and dots characterized by the same colour represent the data distribution per each participant.

## Data Availability

Research data may be shared after a specific request to the corresponding author.

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
