# Peer review of "Wearable Technologies for Electrodermal and Cardiac Activity Measurements: A Comparison between Fitbit Sense, Empatica E4 and Shimmer GSR3+"

_sensors, 2023, doi:10.3390/s23135847_

Round 1

Reviewer 1 Report

In this study, the author evaluated the reliability of three devices: Fitbit Sense, Empatica E4 wristband, and Shimmer GSR3+. Although the work is relatively straightforward, it presents interesting findings. After addressing the following questions, I would like to recommend accepting this manuscript.

1. Fitbit Sense is already commercially available, is there any point in comparing it? Please make a discussion.

2. What is the purpose of the study and what is it trying to illustrate?

3. The discussion section is redundant. This part should be added to the corresponding results section.

4. The authors had utilized statistical methods for the analysis. However, these methods were not described in the experiments section, please add a detailed description.

5. Please cite some well-respected literature, not niche papers.

Minor editing of English language required

Reviewer 2 Report

The manuscript was well-design and it compared to the commonly used devices in the research. I have only some advice which given below to become better.

1. The refers in the main text should be fixed based on the journal rules such as [1,2]. Also the figure legends should be fixed too. 

2. In method part, it should be given the volunteers’ weight and height information mean values.  If a person has more fat tissue, vessels could be covered more fat. So, it could be affected measurement of HR due to the laser signal could be changed. The information is important for results.

3. In line 189, it could be preferred non-dominant instead of no-dominant.

4. In the material method, it should be mentioned that how chosen participants number in the groups. Even if, g power analysis should be given.

5. In line 245-246, the writing in the parenthesis should be fixed.

In line 189, it could be preferred non-dominant instead of no-dominant.

Reviewer 3 Report

Authors have presented an article entitled “Wearable Technologies for Electrodermal and Cardiac Activity measurements: a Comparison between Fitbit Sense, Empatica E4 and Shimmer GSR3+” Though the manuscript is well written and organized but there is scope for further improving the quality of the draft before considering for publication.  

Few minor comments are listed below:

1. As the author mentioned that wearable technologies are growing in terms of popularity for monitoring health-related features. Author should send the performance of related articles to make it contrast: Fibers and Polymers 20, 1161-1171 (2019).

2. In Figure 3, author should incorporate the captions of red and green contained arrows within the figure.

3. Figure 3 needs to be numbered as (a) and (b) for left and right bars.

4. Figure 4 needs to be corrected the same as previous.

5. Authors should check the typos and grammatical mistakes in the revised version. 

They need to check the typos and grammatical mistakes in the revised version
